# SecureLLM: Using Compositionality to Build Provably Secure Language Models for Private, Sensitive, and Secret Data

## Abstract

As Large Language Models (LLMs) increasingly support critical sectors such as healthcare, finance, and public governance, ensuring data confidentiality and robust access control is a pressing societal challenge. Traditional security mechanisms isolate sensitive resources from unauthorized users, yet existing LLM safety approaches often fail to enforce strict segregation of confidential data. In this work, we introduce *SecureLLM*, a novel compositional framework for building provably secure large language models (LLMs) that integrates fine-tuning with traditional access security measures to protect private information. By fine-tuning LLMs on segregated, "siloed" training data and composing their outputs at inference time based solely on a user's verified credentials, SecureLLM not only prevents unauthorized data leakage but also enables accurate responses for complex queries spanning multiple data silos. Our method is demonstrated on a challenging natural-language-to-SQL translation task and is designed with real-world applications in mind—supporting sectors where protecting sensitive information is paramount.

## 1 Introduction

In today's data-driven society, the deployment of large language models (LLMs) in domains like healthcare, finance, and public administration promises transformative benefits but also introduces significant risks. While LLMs excel at natural language understanding and generation, their vulnerability to prompt injection and data leakage can compromise sensitive information—a risk that is unacceptable in applications that must adhere to strict privacy and regulatory standards.

LLMs can convinced to reveal sensitive information as is demonstrated by popular prompt hacking techniques for malicious content generation like in Do Anything Now (DAN) (Shen et al., 2024). The use of "Guardrails" models to detect malicious generation has been a sufficient method for online models to sanitize outputs before presenting them to the user, but for every guardrail developed, there is always another method developed shortly thereafter to break said guardrail (Mangaokar et al., 2024; Banerjee et al., 2024; Dutta et al., 2024; Andriushchenko et al., 2024). For enterprises where security must be guaranteed by local laws and regulations, like finance, healthcare, and national security, guardrails are not legally sufficient to prevent leaking sensitive information. No prior work offers a method that provides a guarantee of data security for information silos that must be stored separately and maintain credential-based access controls, which severely limits LLM adoption in security-focused fields. We provide the first method to build provably secure LLMs by reflecting the compositionality that allows LLMs to be as secure as credential-based security.

Global initiatives such as the United Nations Sustainable Development Goals (SDGs) and the Leave No One Behind Principle emphasize the urgent need to protect individual privacy and ensure equitable access to technology. Motivated by these societal imperatives, our work rethinks LLM security by directly embedding traditional access control mechanisms into the model architecture. We introduce *SecureLLM*, a framework that leverages the compositional nature of security by fine-tuning separate LLMs on isolated data silos and dynamically composing their outputs at inference time according to user-specific permissions.

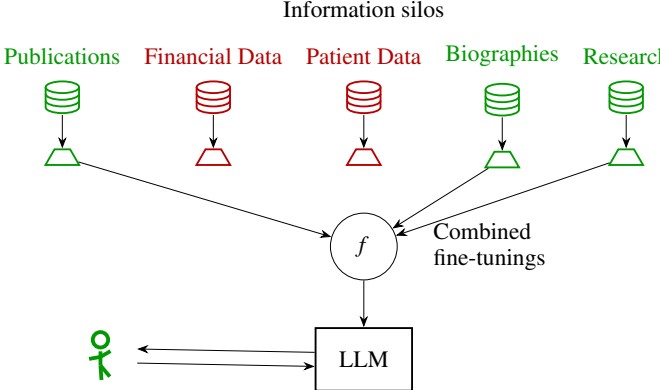

Figure 1: Assuming a perfect compositional function $f$ that runs at inference time, we propose a method that guarantees information security. Each model is fine-tuned on a previously segregated information silo. The user's credentials are validated using traditional security methods, and inference is only run on models for which the user has verified access. The outputs of each fine-tuned model are composed at inference time with the function $f$ and that single composition is passed to the user. Thus, SecureLLM reduces the problem to LLM security to that of existing information security systems. Existing compositional fine-tuning methods fail in this challenging environment. SecureLLM presents a new method that better approximates the function $f$.

This design not only guarantees that users receive responses only from data to which they are authorized but also enables accurate cross-silo queries that no single silo can resolve independently. In addressing the dual challenges of robust data security and high-performance generalization, SecureLLM directly contributes to the broader mission of AI for Social Good—ensuring that advanced AI technologies are both ethically deployed and beneficial to society. Moreover, our approach encourages collaboration across multiple disciplines, inviting contributions from computer scientists, security experts, and domain stakeholders to collectively safeguard sensitive information in real-world applications.

We consider the scenario where an organization has a set, $N$, containing separate and confidential data silos that must be kept separate for legal purposes, but there are also users who have access to some arbitrary subset of $N$. We make the following assertions of properties that must be present to call a model secure:

1. Can accurately respond to prompts on data that the user already has verified access-credentials
2. Can accurately respond to prompts that require the intersection of segregated data silos
3. Will *provably never* reveal information to an unauthorized user

Trivially, one could fine-tune many models on the power set of $N$; but this has a major flaw. Using this trivial method, the number of models required to satisfy our Secure Model Properties is $2^n$, or $2^n - 1$ if we reasonably do not consider the empty set. This quickly becomes impractical for values of $n > 4$. Instead, we show how to achieve the same goals with a linear number of LLM fine-tunings (fig. 1). While using only one fine tuned model per silo, we can configure and compose a model specific to the user's permissions at runtime.

While other methods have demonstrated compositionality for similar tasks, there are none that have been designed for situations where information silos are entirely orthogonal and disjoint from one another. To rigorous demonstrate the compositional properties of SecureLLM, we formulate a new compositional task using natural-language-to-SQL translation. In this task, each SQL schema is entirely disjoint and prompts do not contain the exact table or column name, thus requiring the model to have perfect parameterized knowledge of the schema. SQL translation offers an extreme test of compositionality, and only serves to demonstrate in an easily verifiable manner the efficacy of SecureLLM compared to other compositional methods. For practical SQL translation of the same task, it is simply easier to pass the siloed database schemas as part of the prompt to achieve the same result.

This idea is reminiscent of recent work like LoraHub (Huang et al., 2023) which also composes fine-tunings. Given a target task, LoraHub selects a set of fine-tunings, Low Rank Adapters (LoRAs) (Hu et al., 2021), that are added together. However, LoraHub is designed for soft tasks, where a model already tends to perform well, and where LoraHub's aim is to increase performance by a few percent. The domain we consider here is radically different, we seek compositionality for unrelated silos of information. Questions that require access to silos A and B, by definition cannot be answered with access to only A or only B. The underlying performance of models is nearly zero. LoraHub is not well-suited to such tasks and performs very poorly. Our compositional method in SecureLLM solves this issue and can generate accurate responses when the intersection of silos A and B is required.

Our contributions are:

1. formulating a difficult new compositional task that LLMs have great difficulty with – natural-language-to-SQL where not only a model is trained on queries of individual databases but also where the accurate responses require cross-database joins,
2. formulating the notion of access security in terms of this task,
3. demonstrating that existing fine-tuning methods fail in this compositional environment,
4. introduction of new compositional methods for this problem.

While we only concern ourselves with the task of understanding queries by translating them to SQL, we claim our methods are generic and can be applied to numerous other domains like translating commands to API calls and answering questions from large collections of documents.

## 2 RELATED WORK

**Model Composition**. Our framework relies on composing LLM fine-tunings at inference time which follows a set of previous works that use model composition. A recent method combines pretrained LLM prompts each tuned for separate tasks to achieve generalization on downstream tasks (Sun et al., 2023) but requires training prior to inference. AdapterSoup composes fine-tunings by linearly averaging fine-tuned weights depending on a criteria to determine which fine-tunings are relevant to the new domain (Chronopoulou et al., 2023). PEM Addition is a method that doesn't require further training such as composing fine-tunings using arithmetic operations directly on the weights (Zhang et al., 2023). LoraHub is a recent simple framework that also composes different LoRA fine-tunings (Hu et al., 2021) at inference time where each fine-tuning is trained on a different task (Huang et al., 2023); we include comparisons using LoraHub and PEM Addition.

**Privacy Attacks**. Many recent works discuss a range of different privacy attacks against large language models and Deep Learning models in general. Membership inference attacks are a type of privacy attack which try to determine if a piece of text was contained in the training data of a model possibly without access to the weights (Hisamoto et al., 2020; Nasr et al., 2019; Hu et al., 2022). An even larger security risk is posed by training data extraction attacks where large language models leak text in their training data verbatim (Carlini et al., 2019) including personally identifiable information (Inan et al., 2021). This attack is shown to be successful even when such data was only mentioned in a single document and this behavior worsens with an increase in model size (Carlini et al., 2021). Similarly, training data extract attacks were effective on models fine-tuned on a smaller dataset (Zanella-Béguelin et al., 2020). With recent work tackling these privacy issues for Retrieval-Augmented Generation using multi party communication (Zyskind et al., 2023).

**Differential Privacy**. A popular algorithmic technique to train machine learning models with certain privacy guarantees is differential privacy (Abadi et al., 2016) which has also been applied to large recurrent language models (McMahan et al., 2017). Multiple recent works manage to use differentially private learning on large language models with hundreds of millions of parameters to achieve efficient differentially private fine-tuning with slight degradation in performance (Li et al., 2021; Yu et al., 2021). Many other methods borrow inspiration from differential privacy like Confidentially Redacted Training which provably prevents memorization of the training data (Zhao et al., 2022). However, there are differences between Differential Privacy and our approach. In differential privacy, there exists a non-zero amount of privacy loss parameterized by the privacy budget ($\epsilon$ and $\delta$) from the resulting model as the privatization step minimizes but does not completely ensure that the updated model parameters do not leak private information.

## 3 FRAMEWORK

SecureLLM takes several fine-tunings each trained on distinct information silos and composes them at inference time. The goal of the composed model is to answer questions about both individual silos and questions that span silos. For example, in our case, a natural-language to SQL LLM would need to be able to generate joins across the databases of multiple silos to answer complex questions that have never been seen at training time. This is a trivial task for humans, but one that challenges LLMs. We go a step further: not only must such an LLM work, it must operate through a combination of fine-tunings, i.e., not only has it never seen combinations of silos at training time, its fine tunings have only ever seen a single silo each. This challenges, and defeats, current fine-tuning methods. The upshot of this difficult task is that it solves several key security problems for LLMs.

Given $N$ data silos $\{S_1, S_2, \cdots, S_N\}$ and $N$ fine-tuned LLMs $\{M_1, M_2, \cdots, M_N\}$ where $M_i$ has been fine-tuned on the data silo $S_i$, and given a set of target indices $T \subseteq \{1, 2, \cdots, N\}$, the goal is to obtain a composed model $M_T := M_{T_1} \oplus \cdots \oplus M_{T_{|T|}}$ at inference time with no additional training such that $M_T$ is able to correctly answer any question about the information contained in the target silos $S_i, \forall i \in T$ and should fail to answer any question about information not contained in the target silos $S_j, \forall j \notin T$ as to not leak any information that the desired model $M_T$ is not intended to have. Additionally, the target model $M_T$ should be able to answer new *union questions* $q_{union,ij} \in S_{i \cup j}$ where $i \in T \wedge j \in T$ where the question relies on information contained in both $S_i$ and $S_j$. We note that the union questions $q_{union,ij}$ are not answerable by any individual data silos, thus none of the individual models $M_i$ are able to answer any union questions while a successfully composed model should be able to answer such questions without the need of any training.

It is critical that the composed model $M_T$ has no knowledge of information silo that the user is not authorized to access, i.e. data silos $S_i, i \notin T$. Without this condition, a trivial solution is to train a single model $M_{All}$ on all data silos $\{1, \cdots, N\}$ however this approach is susceptible to leaking confidential information as the model would have knowledge of information contained in silos that users are not authorized to view and thus is not a valid approach. This approach is also problematic for scenarios that employ security through contradiction, in that some silos may directly contradict information in another silo in order to protect sensitive information (SecureLLM could potentially solve this by applying weights to silos of higher confidentiality). We refer to $M_{All}$ as the Exponential Model that has seen every combination and such a model is used as an insecure upper bound to performance in our experiments.

An alternative to composing fine-tunings while also preserving privacy would be to create an exponential number of models, one for the powerset of information silos. This would maximize performance and minimize the amount of generalization needed, as long as one had a way to automatically generate cross-silo questions, perhaps with another LLM. This is obviously impractical. In essence, our method provides the advantages of the exponential approach but with linear storage and training runtime.

### 3.1 COMPOSING FINE-TUNINGS

We discuss several existing methods, none of which perform well. A few other plausible methods that also do not perform well are shown in the appendix. Finally we describe two new methods that do perform well with one clear winner.

**LoraHub** The LoraHub method for composition introduced by Huang et al. (2023) is a two-step process involving element-wise summation of LoRA fine-tunings (*COMPOSE*), and then learning weight optimizations via gradient-free methods to apply to each fine-tuning (*ADAPT*). For this paper, we do not implement the *ADAPT* stage because weights for every possible combination would need to be learned, and we could not say that this process is completed at inference time.

We observe that LoraHub performs poorly on the secure composition task. The authors warn that combining too many fine-tunings can lead to poor performance, however this cannot be the source of poor performance as we compose only up to three LoRAs.

**PEM Addition** The summation method introduced by Zhang et al. (2023) is similar to LoraHub, however, instead of summing the embeddings of the encoder and decoder prior to receiving the input

Q: What's the average age of all teachers that are older than 72 or that taught art classes for 9th graders in the school. Answer:

```
1    SELECT AVG(instructors.teacher_age)
2    FROM instructors INNER JOIN classes
3        ON instructors.teacher_id =
                classes.teacher_id
4    WHERE instructors.teacher_age >= 72
5        OR classes.class_subject = 'art' AND
                classes.level = 9
```

(a) Sample from Silo 1 ($S_1$)

Q: What's the minimum height of all appliances in the inventory that are currently unavailable in stores located in NY, CA, or MA and with a rating higher than or equal to 2 stars. Answer:

```
1    SELECT MIN(inventory.height)
2    FROM inventory INNER JOIN store ON
3        store.store_id = inventory.store_id
4    WHERE inventory.available = 0
5        AND (store.location = 'NY'
6        OR store.location = 'CA'
7        OR store.location = 'MA')
8        AND store.star_rating >= 2
```

(b) Sample from Silo 2 ($S_2$)

Q: Provide the names of all managers located in TX and the names of all teachers that are younger than 86 and that taught english, sociology, or art classes that achieved a grade higher than 89 in the database. Answer:

```
1    SELECT store.name
2    FROM classes
3    INNER JOIN instructors ON
            instructors.teacher_id =
            classes.teacher_id
4    INNER JOIN store ON store.name =
            instructors.name
5    WHERE store.location = 'TX'
6    AND instructors.teacher_age <= 86
7    AND (classes.class_subject = 'english'
            OR classes.class_subject =
            'sociology' OR
            classes.class_subject = 'art')
8    AND classes.grade >= 89
```

(c) Sample from Union Silo 1,2 ($S_{1 \cup 2}$)

Figure 2: Examples of input/output pairs of a question paired with the target SQL query which are unconditional samples from a Context-Free Grammar.

Q: What is the average age of instructors who are aged 72 or older or teach art at level 9? Answer:

```
1    SELECT AVG(instructors.teacher_age)
2    FROM instructors INNER JOIN classes
3        ON instructors.teacher_id =
                classes.teacher_id
4    WHERE instructors.teacher_age >= 72
5        OR classes.class_subject = 'art' AND
                classes.level = 9
```

(a) Sample from Silo 1 ($S_1$) that
was rephrased with ChatGPT

Q: What's the minimum height of all appliances in the inventory that are currently unavailable in stores located in NY, CA, or MA and with a rating higher than or equal to 2 stars. Answer:

```
1    SELECT MIN(inventory.sloth)
2    FROM inventory INNER JOIN store ON
3        store.bear = inventory.bear
4    WHERE inventory.pony = 0
5        AND (store.alpaca = 'NY'
6        OR store.alpaca = 'CA'
7        OR store.alpaca = 'MA')
8        AND store.raccoon >= 2
```

(b) Sample from Silo 2 ($S_2$)
with obfuscated column names

Figure 3: Examples of input/output pairs of a question paired with the target SQL query which (a) are from the ChatGPT rephrased silos and (b) use column names obfuscated by an arbitrary but consistent mapping.

$x$, one executes each fine-tuning independently at the attention-layer level, and then adds the result. This version of summed composition shows improved performance over LoraHub.

**Average of Adapter Weights** Computing the simple average of each Lora fine-tuning response, as suggested by Chronopoulou et al. (2023), $\sum_{i \to n} \frac{L_i}{n}$, produced compositions that were 50% less effective than PEM Addition in initial informal tests.

**Variations of LogSumExp of Adapter Weights** Du et al. (2020) proposes a disjunctive composition process based on Energy Based Modeling, $-logsumexp(-E_1(x), -E_2(x), \cdots)$. Every variation tried performed significantly worse than PEM Addition, and upon closer inspection, this process substantially distorts encoder and decoder embeddings.

**Adapter Concatenation** The Mangrulkar et al. (2022) library implements weight concatenation, however we found that concatenating LoRA encoder/decoder fine-tunings performed significantly worse than PEM Addition.

## 3.2 OUR METHODS

**Maximum Difference** The intuition behind this method is to select the embeddings from each fine-tuning with the strongest response (either positive or negative) at each attention layer. In order to accomplish this, each LoRA fine-tuning is evaluated separately on input $x$. Then a mask of zeros with the same dimension as the output is created, $h_{max}$, to aggregate LoRA responses. For each LoRA fine-tuning response $L_i$, an element-wise comparison is made, and if the absolute values of the fine-tuning response is greater than the aggregated response, then the signed response from that fine-tuning replaces the element in the aggregated response.

**Logit Composition** Given fine-tunings to compose $M_1, \cdots, M_n$ and input $x$, we define logit composition as performing the complete forward pass for each fine-tuning independently to obtain logit probabilities. We select the maximum value of each logit. One could instead sum logits for each fine-tuning. We found little difference between the two implementations, although the sum may have issues as the number of fine-tunings increases.

Note that we are not claiming this method to be a superior compositional approach in every case. The requirements of compositional security are different than those of some other compositional tasks. By its very nature, compositional security implies that most of the time every model but one is irrelevant and confused, and unconfused model is most likely to produce confident results. This motivates our compositional methods and explains why other methods preform so poorly.

## 4 DATA GENERATION

Our goal is to automatically create a challenging dataset for compositions of silos. While there are countless other NL2SQL datasets, none specifically focus on SQL queries for disjoint and unrelated databases silos. Secure-NL2SQL contains three silos of disjoint schemas pertaining to different subjects, as well as the superset of unions between those three silos for a total of seven permutations $(S_1, S_2, S_3, S_{1 \cup 2}, S_{1 \cup 3}, S_{2 \cup 3}, S_{1 \cup 2 \cup 3})$. The dataset contains automatically generated questions and corresponding SQL queries across each silos.

We automatically generate SQL databases, one per silo, with 2-3 tables per database, that share columns which can be joined together both within and between databases. However, the databases are otherwise disjoint and contain different topics. For each database we generate natural language questions along their equivalent SQL. Then, we generate questions and SQL pairs that span pairs and triples of databases. Two methods are used to generate these pairs: a CFG (see fig. 2) and ChatGPT 4 (see fig. 3). The CFG generates both the SQL and the question in parallel. We do this at large scale, with 100,000 pairs per silo or combination of silos. To ensure that our results scale to more realistic queries we also generate 300 pairs per silo or combination of silos.

We limit the scope of generated SQL statements. All statements generated from our CFG, an excerpt of which is shown in the appendix, are SELECT statements that only contain the SQL keywords FROM, NATURAL JOIN, and WHERE. The majority of the complexity is in the WHERE clause which requires specialized knowledge about the schema along with language comprehension to properly generate based on the input question. The task for the LLM is to generate the WHERE clause of an SQL statement which answers the input question.

We introduce a useful normalization, for which we provide ablations in the results section. This normalization is closely related to 6NF (Date et al., 2003) by ensuring table and columns are not "guessable" by a non-fine-tuned model based on context provided in the prompt. In general, this transformation could help all SQL LLMs, and is bidirectional. As described in the results section, our composition methods are far superior irrespective of this normalization, but we believe it is a valuable observation that is likely to lead to many more LLM-specific normalizations as they become serious consumers of SQL.

## 5 EXPERIMENTS

To demonstrate the capabilities of model composition at inference-time, we first begin by obtaining individual fine-tunings that are knowledgeable in a single silo by fine-tuning a Llama-2-7b model for each silo separately. The fine-tuning results in a Low-Rank Adaptation (LoRA) for each silo which can independently be applied to the base Llama-2-7b model. Once the individual LoRA fine-tunings are obtained we compose them using one of several compositional schemes with the requirement that the composition happens at inference time with no additional training. We additionally train two insecure baseline models that act as an upper-bound using LoRA, the baseline generalized model is trained on all the individual silos together and must then generalize it's knowledge to the union silos. While the baseline exponential model also breaks privacy guarantees by training on all the individual silos along with the union silos, the term exponential refers to the fact that training such a model while preserving privacy would mean that $\mathcal{O}(2^N)$ models would need to be trained where $N$ is the number of silos in the database. Both baseline models are considered insecure as there is

Table 1: Normalized tree edit distance for CFG-generated question and SQL pairs with accuracy reported in parentheses (average and std. dev. only applies to normalized tree edit distance). The exponential baseline sees all combinations of silos at training time, this is intractable and insecure, but has maximal performance. The generalization baseline sees all silos but not combinations of silos at training time, this is tractable but insecure. The other methods are used to build a SecureLLM. As described above, we do not include detailed reports on methods which underperform both LoraHub and PEM Addition. Note that our methods significantly outperform prior work. They retain all the generalization performance there is (since the generalization model sees all silos at once, while the fine-tunings each see silos separately, the generalization model should nominally perform better), even outperforming the generalization baseline.

| CFG Generated | Baseline Exponential Model | | Baseline Generalized Model | | LoraHub | PEM Addition | Ours (Maximum Difference) | Ours (Logits) |
|---|---|---|---|---|---|---|---|---|
| $Silos_1$ | 0.0 | (100.0%) | 0.0 | (98.3%) | 1.9 | 0.9 | 0.4 | **0.1** |
| $Silos_2$ | 0.0 | (96.7%) | 0.0 | (100.0%) | 2.6 | 0.8 | 0.3 | **0.1** |
| $Silos_3$ | 0.0 | (100.0%) | 0.0 | (100.0%) | 1.2 | 0.7 | 0.2 | **0.1** |
| $Silos_{1\cup2}$ | 0.0 | (99.2%) | 0.5 | (0.0%) | 1.7 | 0.7 | 0.7 | **0.2** |
| $Silos_{1\cup3}$ | 0.0 | (100.0%) | 0.4 | (1.7%) | 2.0 | 0.7 | 0.6 | **0.3** |
| $Silos_{2\cup3}$ | 0.0 | (100.0%) | 0.5 | (1.7%) | 2.4 | 0.7 | 0.7 | **0.2** |
| $Silos_{1\cup2\cup3}$ | 0.0 | (98.3%) | 1.0 | (0.0%) | 1.8 | 1.0 | 0.9 | **0.2** |
| $\mu \pm \sigma$ | $0.0 \pm 0.0$ | | $0.35 \pm 0.38$ | | $1.95 \pm 0.47$ | $0.78 \pm 0.15$ | $0.56 \pm 0.26$ | **$0.19 \pm 0.1$** |

no method of removing knowledge about certain silos at inference time when the user does not have the sufficient credentials unlike our proposed SecureLLM method which can remove and add fine-tunings with each silo's knowledge at inference time. We fine-tune all models with one epoch until saturation (achieving near 100% accuracy on the CFG validation set) using a frozen Llama-2 7B (Touvron et al., 2023) with a trainable LoRA fine-tuning (Hu et al., 2021) using LoRA parameters $r = 8, \alpha = 32$ and a dropout (Srivastava et al., 2014) of 0.1, an Adam optimizer (Kingma & Ba, 2014) with a learning rate of 0.0002, a batch size of 32, and a weight decay of 0.002.

While Exact Match (EM) accuracy is typically recorded for NL2SQL datasets, we found this metric to not be granular enough to show differences in method performance. Instead, we calculate the tree-edit distance Zhang et al. (1996) between the ground query and the generated query. By computing the number of edit operations required to transition between the two, we can show how close a given generated query is to the correct query, whereas using only EM is a binary representation of correctness.

We report the results of the two insecure baseline models along with the secure composition ($M_1 \oplus M_2 \oplus M_3$ where $M_i$ was trained on $Silo_i$) using multiple compositional methods (as described in Framework) including our best method (Framework: Logit Composition) with and without using 6NF-like database normalization, which is equivalent to the scenario where a user has credentials to access $T = \{1, 2, 3\}$. We note that neither the baseline generalized model nor the secure compositions have seen the union Silos ($S_{1\cup2}$, $S_{1\cup3}$, $S_{2\cup3}$, and $S_{1\cup2\cup3}$) and that only the exponential

Table 2: Results on the ChatGPT-paraphrased questions. See Table 1 for a detailed explanation. Our method continues to outperform all others, and again outperforms the generalization baseline. Scaling to realistic queries still favours our approach.

| GPT Generated | Baseline Exponential Model | | Baseline Generalized Model | | LoraHub | PEM Addition | Ours (Maximum Difference) | Ours (Logits) |
|---|---|---|---|---|---|---|---|---|
| $Silos_1$ | 0.0 | (87.5%) | 0.1 | (79.2%) | 2.0 | 1.1 | 0.5 | **0.2** |
| $Silos_2$ | 0.2 | (61.7%) | 0.2 | (56.7%) | 2.8 | 1.0 | 0.5 | **0.3** |
| $Silos_3$ | 0.1 | (56.7%) | 0.2 | (51.7%) | 1.4 | 1.1 | 0.5 | **0.2** |
| $Silos_{1\cup2}$ | 0.2 | (29.2%) | 0.4 | (0.0%) | 1.5 | 0.9 | 0.6 | **0.4** |
| $Silos_{1\cup3}$ | 0.1 | (33.3%) | 0.3 | (3.3%) | 2.2 | 0.6 | 0.5 | **0.3** |
| $Silos_{2\cup3}$ | 0.1 | (50.0%) | 0.3 | (2.5%) | 1.9 | 0.6 | 0.5 | **0.2** |
| $Silos_{1\cup2\cup3}$ | 0.2 | (20.8%) | 0.4 | (0.0%) | 2.0 | 0.7 | 0.6 | **0.2** |
| $\mu \pm \sigma$ | $0.15 \pm 0.07$ | | $0.28 \pm 0.14$ | | $1.96 \pm 0.47$ | $0.84 \pm 0.22$ | $0.52 \pm 0.05$ | **$0.27 \pm 0.06$** |

Table 3: Following Table 1 while obfuscating the table names. Our methods continue to perform well showing that they are not taking advantage of a trivial solution. For real-world applications, one would likely use a much larger baseline model. This would improve the absolute execution scores, which would method would benefit from since it retains the performance of the underlying model in this challenging compositional task.

| Obfuscated Generated | Baseline Exponential Model | Baseline Generalized Model | LoraHub | PEM Addition | Ours (Maximum Difference) | **Ours (Logits)** |
|---|---|---|---|---|---|---|
| $Silos_1$ | 0.0 (99.2%) | 0.0 (94.2%) | 1.8 | 1.1 | 0.5 | **0.2** |
| $Silos_2$ | 0.0 (92.5%) | 0.0 (100.0%) | 3.1 | 1.4 | 0.5 | **0.2** |
| $Silos_3$ | 0.0 (100.0%) | 0.0 (100.0%) | 0.9 | 0.8 | 0.5 | **0.1** |
| $Silos_{1\cup2}$ | 0.0 (98.3%) | 0.4 (0.0%) | 1.3 | 1.4 | 0.7 | **0.3** |
| $Silos_{1\cup3}$ | 0.0 (77.5%) | 0.6 (1.7%) | 1.6 | 2.5 | 1.0 | **0.5** |
| $Silos_{2\cup3}$ | 0.0 (100.0%) | 0.4 (1.7%) | 1.9 | 2.5 | 0.9 | **0.3** |
| $Silos_{1\cup2\cup3}$ | 0.0 (80.0%) | 0.7 (0.8%) | 1.6 | 2.2 | 1.1 | **0.5** |
| $\mu \pm \sigma$ | $0.01 \pm 0.01$ | $0.31 \pm 0.3$ | $1.73 \pm 0.69$ | $1.7 \pm 0.7$ | $0.73 \pm 0.26$ | **$0.29 \pm 0.15$** |

baseline model has been trained on those silos. The performance of the composed fine-tunings on the individual Silos would give an indication as to whether the resulting composition is able to retain the knowledge of each individual fine-tuning from each separate Silo; This performance is expected to be traded off for privacy while the better compositional methods mitigate the extent of this trade off and maintain maximal privacy. The performance on the union Silos indicates whether the composed fine-tunings are able to successfully generalize knowledge from the individual fine-tunings which is an essential component in answering questions that no individual fine-tuning or silo can answer.

# 6 RESULTS

The highest performance one can possibly achieve is if the LLM is trained not just on every silo but the powerset of silos, i.e., the insecure baseline exponential model described above. Realistically, our model is upper-bounded by a variant of the model that sees all of the silos at training time, but sees no combination of silos. Both of these are insecure, in that they have access to all of the data. Our goal is to find a method to combine individual silo fine-tunings to reproduce the performance of the baseline generalized model.

Raw overall performance is not a relevant metric, although we report it in each case for the baseline models. Raw performance is a function of the size of the model. And we use the modestly sized Llama-2-7b. What is critical is the fraction of retained performance. This is what we focus our results on, the difference in tree edit distance.

LoraHub and PEM addition were the only two competitive methods that were previously published. All other methods described earlier performed so poorly we did not include them in the final results table to make room for additional experiments.

In Table 1 we report performance for the CFG-generated data. Note that for every probe silo combination our methods have by far the lowest tree edit distances. Even without the database normalization described above, our methods outperform all others in every case. With the database normalization our method retains all the performance that exists, i.e., it nearly always matches the tree edit distance of the baseline generalized model.

One might wonder if these results are merely an artifact of the CFG-based approach. When replicating the same experiment with sentences rephrased by ChatGPT, see Table 2, we come to the same conclusions. LoraHub and PEM Addition, along with all prior methods we attempted significantly underperform our approach. Note that this is an extremely challenging test set as the ChatGPT paraphrases are only used for testing, not for training.

To guard against a potential trivial solution to this problem, we also introduce a column-name obfuscated version. A model that is good at guessing a likely name for a table based on the entities it

refers to might otherwise get a leg up. We are interested in the ability of models to retain compositional reasoning, rather than circumventing the task. In any case, in real world conditions column names are often rather complex. In Table 3 each column is given an arbitrary but stable and coherent name, in this case an animal. Relative to Table 1 our method loses little performance, meaning that it encourages compositional reasoning.

# 7 CONCLUSION

Ensuring the secure deployment of large language models is critical for their use in sensitive and societally important applications. In this work, we introduced *SecureLLM*, a novel compositional approach that integrates traditional access security with fine-tuning techniques to construct provably secure LLMs. Our framework effectively mitigates data leakage and prompt injection risks while retaining the model's ability to generalize—demonstrated on a challenging natural-language-to-SQL translation task.

By embedding security directly into the LLM's architecture, SecureLLM offers a scalable solution for sectors such as healthcare, finance, and public governance, where the protection of sensitive data is non-negotiable. Beyond its technical contributions, our work aligns with global initiatives aimed at safeguarding individual privacy and promoting ethical AI. It serves as a blueprint for interdisciplinary collaboration among AI researchers, security practitioners, and policy makers dedicated to advancing AI for social good.

Looking forward, future work will extend these methods to additional application domains—such as secure document question-answering and multi-modal data processing—and will deepen collaborations with governmental and nonprofit organizations. One possible followup could look at the inverse task, given a question determine the silos necessary to answer it. This could be used to monitor conversations to ensure individual privacy remains protected throughout a conversation. Another possible direction would be to look at negative silos that exclude information. A negative silo would explicitly avoid a topic, which would prevent accidental leaks. Models could rewrite text or data to refer or exclude particular silos. The traditional world of access security is rich with problems for LLMs to address, and our work opens up the way for doing so. In addition, by providing provable security, we take a key step toward enabling the use of LLMs in secure environments. In doing so, we aim to further bridge the gap between cutting-edge AI research and real-world societal impact, ensuring that the benefits of AI are harnessed safely and equitably.

**Limitations** We disentangle and address a very specific slice of LLM safety, one that is often commingled with a larger story about safety. SecureLLM only concerns itself with security in the traditional sense: quantized access permissions to data. It relies on traditional security techniques to manage access permissions. There is a widely held belief that LLM security is a totally disjoint new field, but as we show, with the SecureLLM approach we can reduce many of those security problems back to traditional access permission issues. Many security problems of LLM are manageable through traditional means when one can assume that only vetted actors have access, just as they are with current document storage systems. The same systems which ensure patient privacy, financial privacy, and that manage secret information today can be used to manage collections of fine-tunings, and the same supervision methods can trace access to the LLM.

From this perspective SecureLLM solves data leakage and prompt injection attacks, in the same sense that traditional security solves data leakage: those without permissions cannot access this information, and those with permissions have access with training and supervision. Although, in the future one might imagine extensions that provide more fine-grained permissions. Organizations are already set up for this form of security, both for managing the user permissions and for the associated documents, making the deployment of SecureLLM straightforward. In settings where this structure is not available or not appropriate, SecureLLM is not directly applicable, but those scenarios never had meaningful security to begin with.

Explicitly out of scope are other notions of safety and security. For example, the LLM may still fabricate information, produce toxic or biased results, follow guidance that it should not, etc. The only mitigation that we offer is that only a user that has permissions to that data will be impacted directly; the user will still need appropriate training about the limitations and dangers of LLMs.

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

## A    REPRODUCIBILITY

We use NVIDIA Titan RTX 24GB VRAM GPUs for all our experiments. For all of our PEFT parameters, less than one GPU-hours per PEFT was required for training. Between around one GPU-hours was required for composition growing with respect to the number of compositions. For

each run, approximately 20 GB of VRAM is needed as we use half precision for all training and inference. Each PEFT can be trained and inferenced on one GPU. We estimate a total of 10 GPU-hours is required to replicate results for training, and 20 GPU-hours is required to perform the same experiments described in this manuscript.

All code and data required to reproduce our work is provided in the online supplement which will be made public under an open source license.

## B USE OF LLMS

An LLM and other writing tools were used to detect grammar mistakes and typos during the last stages of writing.

Additionally, as described in the data generation process, ChatGPT was used to generated the challenging test set for paraphrased question and the results are reported in Table 2.

