# OpenReview forum: "SecureLLM: Using Inference-time Compositionality to Build Secure Language Models for Private, Sensitive, and Secret Data"
_ICLR.cc/2026/Conference — Submitted to ICLR 2026_

### Official Review · Reviewer_jmMA · 2025-10-21

**Soundness:** 2
**Presentation:** 2
**Contribution:** 1
**Rating:** 2
**Confidence:** 5

**Summary:**

To tackle security vulnerability of LLMs, this paper proposes SecureLLM which is a compositional access-control framework. In SecureLLM separate LoRA-tuned models are trained on disjoint siloed datasets and then composed only at inference according to a user’s verified credentials. The main aim of paper is to guarantee that responses cannot incorporate information from unauthorized silos while still answering cross-silo queries. By embedding security directly into the LLM’s architecture, SecureLLM aims to provide a scalable solution for sectors such as healthcare, finance, and public governance.

**Strengths:**

The paper formalizes a security objective tailored to specific settings in which the composed model must answer only from authorized silos and support cross-silo queries without ever training on unauthorized entities. The Logit Composition rule requires no extra training and potentially outperforms LoraHub and PEM Addition on cross-silo NL2SQL.

**Weaknesses:**

While the paper frequently states provably secure, it provides no cryptographic, information-theoretic, or formal security proof that composition prevents leakage beyond credentialed silos. The security argument mainly relies on data isolation at training time and runtime selection, not on formal noninterference properties. Empirically, there is no test of membership-inference or training-data extraction on individual silos or the composed system.
The paper claims several application domains such as healthcare, finance, and public governance can benefit from this study. However, the benchmark is largely synthetic and does not even slightly examine real-world datasets split into silos, nor other tasks that can support realistic ambiguity, distribution shift, and noisy access policies.
The comparisons include LoraHub, PEM Addition, and Adapter averaging, but omit more recent studies on model/adapter merging and collaborative LLMs. It is crucial to atleast provide an expanded empirical position vs. current research studies on merging/ensembling would strengthen claims of novelty.
Overall, this paper aimed to tackle a very interesting problem. However, the solution is not well-founded and is not theoretically supported.

**Questions:**

-Can you specify an explicit threat model (adversary capabilities, query access, collusion across users), a noninterference-style property for authorized vs. unauthorized silos, and a leakage metric (e.g., information flow)? Can you show that, under your inference-time routing and per-silo training, the composed distribution over outputs is independent of unauthorized silos? If not, provide detailed list of assumptions under which your claims hold.
-Can you explore and include real NL2SQL (e.g., Spider) partitioned into legal silos, and evaluate end-to-end execution accuracy?
-Can you include recent model-merging and adapter-collaboration baselines (e.g., task arithmetic variants, sparse/TIES merging, ensemble-with-router methods, and strong baselines recommended in recent studies)?
-Can you provide latency/throughput analyses as the number of silos grows?
-Can you elaborate more on the distinction of used terms “Private, Sensitive, and Secret”? They have different meaning in the security context and have been used without sufficient context.

---

### Official Review · Reviewer_cHke · 2025-10-31

**Soundness:** 3
**Presentation:** 3
**Contribution:** 3
**Rating:** 6
**Confidence:** 2

**Summary:**

The paper proposes the SecureLLM framework, which constructs provably secure large language models through compositional fine-tuning. To enable cross-silo reasoning while preserving privacy, the research team introduced two novel composition methods: Logit Composition and Maximum Difference. Experimental results demonstrate that this approach achieves performance close to insecure baselines on cross-silo query tasks while maintaining strict access control, validating the effectiveness of the compositional security framework.

**Strengths:**

The paper proposes a novel and practical approach to LLM security by combining compositional fine-tuning with access control. The idea of composing silo-specific models at inference is original and well-motivated. The Secure-NL2SQL task is a valuable benchmark for compositional reasoning. Experiments are thorough and demonstrate clear advantages over existing composition methods. The framework is scalable and has high potential impact in privacy-sensitive domains.

**Weaknesses:**

1. The evaluation is limited to NL2SQL, with no validation on broader tasks like QA or dialogue.

2. The claim of "provable security" lacks formal proof or threat modeling. Scalability to larger models and real-time inference costs are not analyzed.

**Questions:**

1. How does Logit Composition handle cases where multiple silos produce high-confidence but conflicting outputs?

2. How does inference time and memory usage scale as the number of accessible silos increases?

---

### Official Review · Reviewer_kiko · 2025-10-31

**Soundness:** 1
**Presentation:** 2
**Contribution:** 1
**Rating:** 2
**Confidence:** 4

**Summary:**

The paper introduces SecureLLM, a compositional framework for building provably secure language models that integrate access control into model design. It fine-tunes separate LLMs on isolated data silos and composes them at inference time based on user credentials, ensuring authorized access while supporting cross-silo queries. Using a natural-language-to-SQL task, the authors show that their proposed Maximum Difference and Logit Composition methods outperform prior approaches like LoRAHub and PEM Addition, achieving strong compositional generalization and data security.

**Strengths:**

1. Studies an interesting and timely problem with reasonable novelty.

2. Designs a challenging NL2SQL task to evaluate compositionality across disjoint data silos.

**Weaknesses:**

1. The proposed methods are highly heuristic with no theoretical or empirical justification for why they should work; even the underlying intuition is unclear.

2. The experimental setup is limited to a single synthetic NL2SQL task and one Llama-2-7B model, lacking real-world relevance and generality.

3. The results are confusing: theoretically, different composition methods should only affect cross-silo performance (e.g., $S_{1∪2}$, $S_{1∪3}$), since each individual silo model is fine-tuned separately. However, Table 1 shows varying results even on single silos ($S_1$, $S_2$, $S_3$), which should remain identical across methods. This inconsistency requires further explanations.

4. The writing quality is poor: (1) Section 3.2’s method descriptions are unclear and could be formalized mathematically; (2) the experimental setup omits important details (See questions below); (3) several typos and presentation issues reduce readability.

Additionally, the paper contains several issues in presentation and tone, with some degree of overstatement. The use of the term “provably” is not appropriate, as the paper does not provide any formal proof or theoretical guarantee of security—it merely demonstrates empirical isolation through compositional inference. Moreover, the claim in the final paragraph that the proposed method is “generic and can be applied to numerous other domains” is overly strong and unsupported, given that no experiments beyond the synthetic NL2SQL task are presented.

**Questions:**

1. In Section 4, please clarify the structure of each database: how many columns do the 2–3 tables have, and what kind of data do they contain? The data generation process for the 100,000 pairs is also unclear—how were these generated, and how do they differ from the 300 “realistic” pairs?

2. In Table 1, why are accuracy values reported only for the first two methods, while the others only show tree edit distance?

3. (Minor) There are several typos, e.g., L37 “LLM can convinced” → “LLM can be convinced”; L320 “it's knowledge” → “its knowledge.”

---

### Official Review · Reviewer_syhN · 2025-10-31

**Soundness:** 2
**Presentation:** 2
**Contribution:** 2
**Rating:** 2
**Confidence:** 4

**Summary:**

The paper proposed SecureLLM, a framework that enables provably secure LLMs by integrating traditional access control with fine-tuned model composition. It allows LLMs trained on separate data silos to be securely combined at inference time based on user credentials, preventing unauthorized data leakage while maintaining accuracy on complex, cross-silo queries.

**Strengths:**

1.	This paper introduces a compositional mechanism that uses fine-tuning LLMs on access-controlled data silos, which are to be combined at inference time to prevent unauthorized data leakage.
2.	Integrates the traditional concept of securing the confidential data in the LLM security domain to protect the privacy of the personally identifiable information in the LLM systems.

**Weaknesses:**

1.	Since the fine-tuning depends on the pre-defined sources, how does it generalize to the other domains that do not exist in the sources?
2.	As lines 77-84 describe its deployments, how can the SecureLLM be employed in real-world scenarios for generalized cases?
3.	As per the paper's position, how does the proposed framework perform against any of the privacy leakage attacks that reveal personal information? The evaluation should have been illustrated under some real scenarios of attacks.
4.	Why was llama-2-7 b chosen to fine-tune? What justifies its use, given that there are many other advanced models after LLaMA-2 with more capabilities?
5.	How does the integration of the proposed method impact the inference time?

**Questions:**

Please follow the Weaknesses.

---

### Meta-Review · Area_Chair_2VGg · 2025-12-11

**Summary:**

The paper aims at building a framework that enables provably secure LLMs. Therefore, it integrates fine-tuning with standard security measures. The reviewers expressed the following major concerns:
1. The paper is heuristic and not principled enough.
2. The method might not generalize.
3. The threat model is not specific enough.
4. There is no formal security proof.

**Reviewer Concerns:**

No rebuttal was provided.

**Reviewer Scores:**

The reviewers' scores have remained unchanged due to a lack in rebuttal.

---

### Decision · Program_Chairs · 2026-01-26

Reject